# Advancements in Sensor Technologies and Control Strategies for Lower-Limb Rehabilitation Exoskeletons: A Comprehensive Review

**DOI:** 10.3390/mi15040489

**Published:** 2024-04-02

**Authors:** Yumeng Yao, Dongqing Shao, Marco Tarabini, Seyed Alireza Moezi, Kun Li, Paola Saccomandi

**Affiliations:** 1School of Mechanical Engineering, University of Shanghai for Science and Technology, Shanghai 200093, China; yu_yao@usst.edu.cn (Y.Y.);; 2Department of Mechanical Engineering, Polytechnic of Milan, 20133 Milano, Italy; 3Department of Mechanical, Industrial and Aerospace Engineering, Concordia University, Montreal, QC H3G 1M8, Canada; 4College of Mechanical and Vehicle Engineering, Chongqing 400044, China; 5State Key Laboratory of Mechanical Transmission for Advanced Equipment, Chongqing University, Chongqing 400044, China

**Keywords:** lower limb exoskeleton, rehabilitation exoskeleton, comprehensive review, rehabilitation robot

## Abstract

Lower-limb rehabilitation exoskeletons offer a transformative approach to enhancing recovery in patients with movement disorders affecting the lower extremities. This comprehensive systematic review delves into the literature on sensor technologies and the control strategies integrated into these exoskeletons, evaluating their capacity to address user needs and scrutinizing their structural designs regarding sensor distribution as well as control algorithms. The review examines various sensing modalities, including electromyography (EMG), force, displacement, and other innovative sensor types, employed in these devices to facilitate accurate and responsive motion control. Furthermore, the review explores the strengths and limitations of a diverse array of lower-limb rehabilitation-exoskeleton designs, highlighting areas of improvement and potential avenues for further development. In addition, the review investigates the latest control algorithms and analysis methods that have been utilized in conjunction with these sensor systems to optimize exoskeleton performance and ensure safe and effective user interactions. By building a deeper understanding of the diverse sensor technologies and monitoring systems, this review aims to contribute to the ongoing advancement of lower-limb rehabilitation exoskeletons, ultimately improving the quality of life for patients with mobility impairments.

## 1. Introduction

Lower-limb movement disorders substantially impact the quality of human life, bringing a considerable burden on both families and society. It has been reported that millions of people across the globe experience difficulties in walking due to such disorders caused by spinal cord injuries, strokes, and other diseases [1]. These may result in the loss of muscle sensation and related complications such as decubitus ulcers, loss of bone density, lower-limb joint contractures, and deep vein thrombosis [2]. According to a study by the World Health Organization, approximately 500,000 individuals suffer from spinal cord injuries annually [3]. Moreover, the incidence of strokes has significantly increased the number of patients with lower-limb movement disorders due to the increasing elderly population [4,5,6]. In addition, factors such as unhealthy lifestyles, extended periods of working in specific positions, traffic accidents, and warfare can heighten the risk of lower-limb muscle damage [7], thus preventing people from having an adequate quality of life.

Rehabilitation training plays a crucial role in helping patients regain muscles’ functions and increase their chances of recovery. Typically, traditional rehabilitation is facilitated by a physical therapist and concentrates on restoring muscles’ functions or averting atrophy in permanently disabled muscles [4,5]. This process entails a progressive restoration of muscle ability through a sequence of stages, ultimately aiming to enable patients to walk independently and without pain [5]. This process necessitates regularly arranging suitable treatments to restore patients’ muscle abilities progressively. The method of regaining muscle-kinetic ability typically consists of three stages. The first stage seeks to enable patients to move their limbs within a specific range without pain, using external forces for assistance. The second stage involves the gradual reduction of auxiliary forces while promoting active-force generation. In the third stage, patients become capable of walking independently and regain the ability to control their direction and balance [5,8]. Most rehabilitation training is repetitive and time-consuming, resulting in relatively high labor costs. The continuing influx of new patients requiring treatments has imposed high demands for rehabilitation-therapist resources each year [9].

In response to the challenges posed by traditional rehabilitation methods, mechanical devices like rehabilitation exoskeletons have been developed to ease the labor-intensive tasks of physical therapists during rehabilitation training [8]. However, current commercial rehabilitation exoskeletons have limitations, such as being bulky and often necessitating a treadmill or a learning-to-walk-cart-shaped device in the treatment room [10]. It is essential to take into account the load on human muscles when using large machines since excessive loading can lead to severe muscle damage [8]. In recent years, considerable developments have been reported in passive or powered lower-limb rehabilitation exoskeletons that are relatively compact and adaptable to the human body, which offers a more promising solution for patients with movement disorders [11]. These devices aim to overcome the limitations of traditional rehabilitation approaches and bulky commercial exoskeletons while providing enhanced mobility and independence for individuals with lower-limb movement disorders. Moreover, this helps boost users’ confidence in independent mobility by offering a significant amount of assistive power.

Exoskeletons can be categorized into two main groups: those designed to achieve partial-body locomotion or independent walking after rehabilitation and those intended to provide additional torques and power to improve the user’s locomotion [7,12]. Furthermore, some advanced, portable, commercial exoskeletons, such as Rewalk and Ekso [13,14], help users travel by precisely adjusting the positions and angles of the exoskeleton joints using intelligent-control strategies [8]. Despite these advancements, lower-limb rehabilitation exoskeletons have not been widely adopted, likely due to their relatively high cost or the fact that these devices have not yet been extensively validated through experiments.

Integrating sensors, processors, and controllers in exoskeleton development is crucial for monitoring and decision making, ensuring efficient and safe operation. These sensors are key in tasks like visual detection for obstacle avoidance, pressure detection for stability maintenance, and capturing body signals for guiding limb movements [9,12]. Sensors are classified, based on structure, into mechanical or non-mechanical, and by material as flexible or rigid. In rehabilitation exoskeletons, high-precision sensors, coupled with specific filtering methods, are vital for accurate and reliable body-signal capture. The design of the human-exoskeleton interface, incorporating flexible materials, is also crucial for user comfort and experience [15].

Optimizing sensor integration and interface design significantly contributes to advancing lower-limb rehabilitation exoskeletons, offering improved solutions for individuals with movement disorders. While clinical studies have shown that rehabilitation exoskeletons can be more effective than traditional rehabilitators, the effectiveness of these exoskeletons for enhancing outward mobility still requires further validation [16,17]. Sensor application enhances motion-control accuracy and response speed in exoskeleton systems, particularly with the use of sensors like electromyography (EMG) and electroencephalogram (EEG).

The rational planning of sensor distribution in an exoskeleton is essential, as it must align with the functional requirements and structural design of the system, influencing its effectiveness [18]. Actuator selection should consider power, response speed, and load capacity to meet the rehabilitation needs of patients [19]. Advanced control algorithms are key to precise control of the exoskeleton system and improving its rehabilitation effectiveness.

Narayan et al. [20] provided a comprehensive review of the development and control strategies of lower-limb exoskeletons, focusing on enhancing the interaction between the user and the exoskeleton. It categorized control strategies into upper-level and lower-level controls and systematically reviewed the control hierarchy, techniques, and methodologies used in recent years. Hsu et al. [16] conducted a review on the effect of a wearable exoskeleton on post-stroke. It found that exoskeleton-assisted training was superior to traditional training in all phases of walking speed, balance, overall mobility, and endurance. Mohebbi et al. [21] identified various subdomains within assistive and rehabilitation robotics, and emphasizes the complexity of designing, controlling, sensing, and optimizing these systems.

However, these previous systematic reviews often have limitations due to the variety of control strategies employed in the development of lower-limb exoskeletons, including assist-as-needed, model-based, non-model-based, intelligent-based, and hybrid methods. Moreover, most studies have primarily focused on electromyography (EMG) and electroencephalogram (EEG) sensors in the design of lower-limb exoskeletons, neglecting a comprehensive review of other sensor types, such as force sensors and inertial measurement units, as well as the importance of user-driven design in control strategies [18,22].

This study aims to provide a comprehensive review on user-centered sensor technologies and control strategies integrated into lower-limb exoskeletons, evaluating their capacity to address user needs and scrutinizing their structural designs regarding sensor distribution as well as the latest control algorithms. The review also examines the effectiveness of various sensing modalities to facilitate accurate and responsive motion control. It concentrates on a systematic classification and description of exoskeleton hardware research, including the use of functional actuation, advanced epidermal electronic sensors and monitoring systems, innovative joint structures, and user-driven controller design. By highlighting the advancements and identifying the strengths and weaknesses of current developments, this study aims to contribute to the enhancement of lower-limb rehabilitation exoskeletons and improve the quality of life for individuals with mobility impairments. The remainder of this study is organized as follows: Section 2 outlines the methodology, Section 3 discusses sensors and monitoring systems, Section 4 examines control methods in lower-limb rehabilitation exoskeletons, and Section 5 identifies knowledge gaps and potential future research directions.

## 2. Methodology

A comprehensive literature review was conducted for publications from 2014 to 2024, utilizing two scholarly databases: ScienceDirect and Web of Science. The search employed the keywords “(lower limb or foot) & (exoskeleton or robot) & rehabilitation & gait assistance”. This initial search yielded a total of 1057 papers. After excluding 396 entries that were conference, meeting, patent, or review materials, 661 papers remained without duplication. Following an in-depth review of the titles, abstracts, and full texts, 394 papers were deemed irrelevant and subsequently omitted.

The retained papers were all in English and focused specifically on the design and control of lower-limb rehabilitation exoskeletons. The selection criteria were twofold: firstly, articles that reported on innovative exoskeleton designs, including unique sensor and motor distributions or aspects of human-robot interaction; and secondly, articles that introduced pioneering control algorithms, such as novel exoskeleton controller designs and techniques for joint angle or gait control. A total of 55 papers met these criteria. Additionally, 42 relevant papers were identified within the references of these articles, leading to a total of 97 papers deemed relevant to the research objective. These papers were categorized into two main groups: one primarily focused on the design aspects of rehabilitation exoskeletons, and the other on the control algorithms of these devices. The methodology for selection and categorization is graphically illustrated in Figure 1.

In the following review, a comprehensive overview of research papers on lower-limb exoskeletons is provided, detailing the types of sensors and monitoring techniques used, the methods of analysis and control, and the aims, results, and limitations of each study. A summary of various studies focused on the design, development, and testing of lower-limb exoskeletons for rehabilitation and gait assistance is also presented in Tables in later sections. The studies utilized different types of sensors, including EMG and EEG sensors, inertial measurement units (IMU), force, pressure, displacement, and optical sensors, to gather data from users and enhance the performance of the exoskeletons. These sensors, along with different control strategies, help achieve specific aims such as personalizing rehabilitation, enhancing walking assistance, and improving the safety and comfort of the devices. Some of the highlighted aims include the quantification of patients’ exercise data, providing personalized rehabilitation services, reducing muscle load, and making walking more efficient. The results of these studies have demonstrated considerable progress in exoskeleton designs, sensors’ integration, and control algorithms, leading to improved exoskeleton functionality and user experiences. However, these studies also highlight certain limitations and areas for future research, such as verifying the stability and feasibility of the prototypes, improving the sensing and control methods, enhancing the balance and torque capacities, and expanding the subject populations to ensure the effectiveness of these devices in clinical and geriatric settings. Overall, the following review provides an overview of the current state of lower-limb-exoskeleton research and the challenges that need to be addressed to improve the performance and applicability of these exoskeletons.

## 3. Sensors and Monitoring Systems for Lower-Limb Exoskeletons

Sensing technologies used in lower-limb exoskeletons can be broadly classified into three main categories: human signal monitoring, kinematic and kinetic measurements, and environmental sensing. Human signal monitoring plays a crucial role in capturing and analyzing the user’s physiological signals. Biopotential sensors, such as EMG and EEG sensors, illustrated in Figure 2, are pivotal in tracking muscle activity and brain signals. They provide critical insights into the user’s physical state, exertion levels, and cognitive processes. This information is instrumental in customizing the exoskeleton’s functionality to suit the individual needs and capabilities of the user. EMG and EEG sensors are among the most thoroughly investigated and implemented monitoring systems in lower-limb-exoskeleton applications.

Almost all researchers designing exoskeletons have incorporated EMG sensors into their systems [18,23]. EMG sensors offer several benefits in lower-limb rehabilitation exoskeletons, including enabling more natural and intuitive control by capturing the user’s muscle-activity signals, providing personalized assistance based on muscle strength and fatigue levels, and optimizing rehabilitation for individual needs. These sensors also allow for real-time feedback to both users and therapists, facilitating immediate adjustments to the rehabilitation program [23]. On the other hand, there are cons to using EMG sensors. Integrating EMG sensors adds complexity to the exoskeleton system, requiring additional hardware and software for signal processing and control, and it can also be affected by external factors such as sweat, motion artifacts, and noise, which can lead to inaccurate readings and affect the performance of the exoskeleton. Furthermore, an excessive number of EMG sensors can increase discomfort for the user and elevate the cost of the exoskeleton [9,18].

Inertial sensors are used to measure the kinematic and kinetic characteristics, as shown in Figure 3, which focus on tracking the exoskeleton and user’s limb positions, movements, and forces. Inertial sensors like accelerometers, gyroscopes, and magnetometers, often combined into IMUs (Figure 3a), are applied to measure orientations, positions, and movements of both the exoskeleton and the user’s limbs. Force sensors, including strain gauges, capacitive, piezoelectric, and piezoresistive types, are used to measure the forces/torques applied to joints and to assess human-robot interactions. Figure 3b shows an example of force sensors. Flexible thin-film pressure sensors have also been developed for measuring contact forces [24]. Displacement sensors, such as linear, angular, and laser displacement sensors (Figure 3c), are utilized to monitor the movement and position of the exoskeleton and the user’s limbs, contributing to the system’s accuracy and control.

Environmental sensors are designed to detect and respond to the exoskeleton’s surroundings, as shown in Figure 4. Environmental sensors like ultrasonic, infrared, or LiDAR sensors can be integrated into lower-limb exoskeletons to assist with obstacle detection, collision avoidance, and navigation. This category of sensors enhances the exoskeleton’s overall safety and adaptability in various environments.

By incorporating three sensor categories, advanced lower-limb-exoskeleton systems become responsive and adaptable, meeting various rehabilitation and mobility needs. For example, combining kinetic and physiological data, particularly from EMG and EEG sensors, significantly enhances exoskeleton functionality and safety. This allows for personalized, controlled experiences, with kinetic data offering insights into external forces and movements, and EMG and EEG providing details on muscle and brain activity. Such integrated systems promote natural, precise movements, and effective rehabilitation, with real-time monitoring ensuring user safety and comfort. Sensor selection is guided by the exoskeleton’s specific requirements, focusing on user control, autonomy level, and task complexity. This review critically examines these sensors, emphasizing their impact on lower-limb-exoskeleton applications.

### 3.1. Human Signal-Monitoring Sensors

Human signal-monitoring sensors are used to capture and analyze the user’s physiological signals. A review of studies employing these sensors is summarized in Table 1, where each is identified by the lead author alone. EMG sensors measure minute electrical signals (muscle action potentials) produced during muscle contractions on the body’s surface, which have found increasing applications across various fields, such as enhancing speech recognition [31], force estimation [32], and refining disease identification accuracy based on specific features [33]. The introduction of EMG sensors has significantly aided physical therapists in evaluating patients’ muscle mobility during rehabilitation and offered a valuable database for subsequent treatment monitoring and quantitative analysis. However, during the measurement process, noise from heartbeats and external vibrations can disrupt the results and impact the system’s accuracy in detecting muscles’ activations. To minimize the influence of noise on the outcomes, EMG sensors are often combined with filtering circuits [33].

The EEG sensing technique has been gaining increased interest for applications in exoskeletons. Unlike the muscles’ electrical signals that necessitate sensor placements in multiple locations, EEG sensors only need to be densely affixed to the head to capture the majority of the body’s intentions. EEG sensors can be classified into invasive, which involve electrodes being placed into specific brain locations, and non-invasive, which require sensitive sensor elements to be fitted onto the scalp to measure physiological electrical signals [34]. Owing to its high temporal sensitivity and safety, EEG technology is predominantly used in the medical field, assisting clinicians in diagnosing specific diseases. The technology also holds considerable potential in intelligent rehabilitation, often referred to as brain-computer interfaces [44].

Patients with conditions such as spinal cord injuries often experience motor-nerve damage and difficulty transmitting neurotransmitters from the brain to the hand and legs. In these cases, EEG sensors can read the brain’s electrical signals and transmit them through peripheral cables to the motion part, stimulating muscle movement through electrical stimulation or assisting movement via external forces. However, EEG signals suffer from poor spatial resolution, mainly due to signal attenuation as they travel through the skull [45]. Moreover, external disturbances like blinking and changes in breathing can introduce artifacts into the EEG signal, affecting measurement accuracy [34]. To address these challenges, various solutions have been proposed, such as replacing wet Ag/AgCl electrodes commonly used in EEG sensors with a polyvinyl alcohol-glycerol-NaCl contact hydrogel and a 3D-printed silver-coated polylactic acid electrode [44], and utilizing more advanced filters to remove interfering signals [45]. 

### 3.2. Kinematic- and Kinetic-Measurement Sensors

Inertial, force, and displacement sensors are widely used to measure kinematic and kinetic characteristics. Table 2 lists the studies employing such sensors to measure the kinematic and kinetic characteristics of human lower-limb rehabilitation devices, together with the primary aim(s) and major findings. IMUs have garnered significant interest due to their cost-effective nature in measuring body motion. An IMU is an electronic device capable of measuring six degrees of freedom movements, including accelerations along the three orthogonal directions and rotational velocities along three directions (via a three-axis gyroscope). The signals for an IMU can be easily processed to determine an object’s positions and orientations in the three-dimensional space. IMUs are also equipped with a magnetometer to measure gravitational forces. Each sensor is utilized to gather data for the three body axes: roll, pitch, and yaw. As IMUs directly measure linear accelerations and rotational velocities, their output is typically superior to the accelerations and velocities derived from time-based position data. However, a notable drawback of current IMU technology is the drift in position level, which may occur due to noise in the integrated signals, causing a gradual divergence from the actual value [46]. To address the drift issue associated with IMUs, various approaches have been proposed, including sensor fusion (incorporating magnetometers and GPS) and model-based techniques such as extended Kalman filters [47].

In order to directly measure the forces applied on joints and to assess human-exoskeleton interactions, force sensors are widely used. These sensors are adept at discerning forces either generated by or exerted on a device. Their readings influence whether an operation should proceed, thus providing a vital safeguard for both the device and its user [68]. Force sensors are thus used for diverse purposes, such as quantifying the forces applied to robotic arm joints, guiding automated floor sweepers to alter their direction upon encountering walls, enabling tactile light controls, and notifying passengers of unsecured seat belts [69,70,71,72]. Force sensors can generally be categorized according to their operating principles, which include the strain, capacitive, piezoelectric, and piezoresistive effects. These sensors are typically lightweight and durable, making them well-suited for rehabilitation exoskeleton devices. However, as rehabilitation devices advance, force sensors are increasingly designed for the moving parts of limbs, such as muscles, to achieve more accurate motion prediction and control [73]. Flexible thin-film capacitive and resistive pressure sensors have also been developed for measuring interface forces, which may find applications in exoskeletons, considering their compact design and low cost [24]. Unlike the previously fixed sensing environments, sensors used for moving parts need to maintain synergy with the user’s limb. As a result, researchers have proposed using flexible force sensors with a gel structure [74] or a multi-sensor approach to gather data about a moving limb [46].

Apart from the above-mentioned IMUs and force sensors, displacement sensors play a vital role in lower-limb exoskeleton applications by providing precise measurements of linear or angular displacements occurring in the exoskeleton’s joints during movement. They can be broadly categorized as linear displacement sensors, angular displacement sensors, laser displacement sensors, and others, depending on the target’s movement being measured [75]. Strain gauges, inductive, and Hall sensors are the primary measurement methods for small displacements, while technologies such as gratings and magnetic grids are mainly used for larger displacements. These sensors allow for accurate and real-time monitoring of the position and movement of the exoskeleton, which are essential for effective control and coordination with the user’s natural motion. However, current displacement sensors have drawbacks such as high power consumption and susceptibility to external interference. Studies have attempted to employ self-powered techniques, such as generating voltage through an inductive coil [76], but these approaches also face limitations [77].

Owing to their affordability, robust and straightforward construction, and easy-to-read data characteristics, mechanical sensors have been a popular choice among the reported studies. The robotic platform, ALICE [9], features advanced mechanical sensors that can gather diverse types of information generated during human motion in real-time. These data, coupled with Quaternion theory, can further help estimate the user’s kinetic parameters, thereby providing a personalized and safer treatment plan. The rehabilitation-exoskeleton design proposed by Wang et al. [78] also uses purely mechanical sensors integrated with a controller (Figure 5a) [78]. These sensors are capable of executing actions without transmitting data to a microprocessor first, which significantly speeds up the response time and eliminates signal delays. However, mechanical sensors can produce errors due to magnetic fields, vibrations, and elastic deformations in motion [78]. Therefore, lower-limb rehabilitation exoskeletons equipped with these sensors are typically used in more stable environments, such as laboratories and clinical settings.

### 3.3. Environmental and Other Types of Sensors

In addition to the previously discussed sensors, namely, EMG, EEG, IMUs, force, and displacement sensors, some studies have employed different environmental and other types of sensors in the lower-limb exoskeletons for various purposes. Some examples include:Optical sensors: These sensors use light to measure the displacement, velocity, or position of objects. Optical encoders or cameras can be utilized for tracking limb movements, providing data for precise control and coordination of the exoskeleton [79].Ultrasonic sensors: By emitting and receiving ultrasonic waves, these sensors can determine the distance to nearby objects, assisting in obstacle detection and avoidance for exoskeleton users [28].Pressure sensors: These sensors measure the force applied over an area and can be incorporated in the exoskeleton’s foot or contact points to estimate ground reaction forces, enabling force estimation, better balance, and stability control [24,48].Tactile sensors: These sensors can detect touch, pressure, or vibrations and can be embedded in areas where the exoskeleton interacts with the user’s body, providing feedback on fit and comfort [80].Temperature sensors: These sensors monitor temperature changes and can be integrated into exoskeleton systems to ensure that motors, batteries, or other components do not overheat during operation [81].

These alternative sensors can be used individually or in combination with the previously mentioned sensors to enhance the functionality, safety, and user experience of lower limb exoskeletons. By integrating various sensor technologies, developers can create more sophisticated and adaptive exoskeleton systems that cater to the specific needs of users in rehabilitation or assistive applications. As an example, polymer optical fiber curvature sensors have been suggested, notable for their high strain limits and fracture toughness (Figure 5b) [46]. These sensors, which have a similar order of magnitude range of motion to the human body, have found wide use in many rehabilitation exoskeletons. Yet, factors like light-source power deviation can cause angular measurement errors, which need to be compensated by other means. Some researchers suggest integrating two inertial measurement units with Kalman filtering and a polymer fiber curvature sensor based on intensity variation. This approach can yield high accuracy and relatively low cost when measuring small angles [46,78].

In summary, sensors and actuators, when judiciously integrated into the designs, can successfully accomplish diverse functionalities such as ensuring the user’s gait stability, creating precise human models, and gathering motion data [46,82]. As discussed, various sensors are required for different measurements, for instance, pressure sensors are required for the detection of limb and plantar pressure and acceleration, inertial sensors for the measurement of human motion trajectory, and clear-vision sensors for the detection of road obstacles. Incorporating a wider range of sensors can enhance the exoskeleton’s ability to emulate human perception, thereby enabling it to respond with appropriate limb movements in real time.

It should be noted that patients cannot solely rely on the exoskeleton device for their rehabilitation and still require guidance from a physical therapist. This need has led to a growing interest in the field of human-exoskeleton interaction and the proposal of remote intelligent-control systems for real-time monitoring [3,83]. These systems enable the rehabilitation devices to be controlled by the user through touch screens or voice communication for predefined movements. Simultaneously, with the user’s consent, the physician can remotely access the system to provide updated rehabilitation therapy codes. Data collected through sensors allow the physician to make timely adjustments, thereby enhancing the promptness and precision of the rehabilitation process.

## 4. Control of the Lower-Limb Rehabilitation Exoskeleton

One of the main aspects of developing a high-performance lower-limb exoskeleton is designing an effective control strategy. Although various control strategies have been investigated for the actuation and control of lower-limb rehabilitation exoskeletons, efforts to improve these strategies continue. These control strategies aim to replicate as closely as possible the actual limb position in trials involving healthy individuals, and subsequently, those with limb injuries [49,50]. Therefore, user perceptions, such as safety and comfort, must be considered in the control-system design, with a particular focus on gait planning and stability during movement. Ideally, a controller should be optimized to consider both aspects, providing a superior user experience [51]. From the perspective of user-friendliness, controlling a lower-limb rehabilitation exoskeleton initially involves replicating limb movement safely before the user dons the device. Initial control stages should safely mimic limb movements, integrating individual user data through muscle electrical sensors for tailored rehabilitation. The planning of gait trajectories is crucial, particularly for outdoor activities, necessitating detailed monitoring of biomechanical factors such as the center of mass and step length. Challenges posed by the external environment call for both structural adjustments and advanced programming to maintain equilibrium. Research has often focused on sagittal-plane balance; however, comprehensive balance control across all body planes, including the coronal plane for foot rotation, is critical to prevent injuries and align the exoskeleton with human joints accurately. A variety of control algorithms, from classical and intelligent to innovative approaches like moment field control and hybrid models, have been developed to improve stability and movement accuracy in lower-limb exoskeletons. Tailoring control to individual user dynamics, such as limb strength recovery, is vital for ensuring comfort and effective rehabilitation. Techniques that amplify user torque and assess stability during physical activities like squatting contribute to enhancing the functionality and safety of exoskeletons in rehabilitative settings.

These developed control strategies can be broadly classified into four main categories: assist-as-needed (AAN), black box (model-free), gray box, and white box (model-based). These classifications reflect the varying degrees of system knowledge and user interaction required to effectively manage and optimize the exoskeleton’s performance. The black-box model indicates no clear relationship between input and output data, while the white-box model shows definite relationships between the inputs and the outputs. The gray-box model lies between these two, indicating a partial relationship between the inputs and outputs [84]. Given that the lower-limb rehabilitation exoskeleton is designed to fit the human body and considering user safety, the system needs to be as close to a white-box model as possible [85]. The controller output thus needs to be supported by corresponding sensor data. However, factors such as sensor hysteresis, creep, joints’ nonlinearities, and external disturbances pose considerable difficulties in realizing a reliable exoskeleton white-box model [86]. Most of the reported control strategies are based on gray-box models involving linear representations of nonlinear elements.

Moreover, exoskeletons require different controllers to achieve specific goals. Sliding-mode control ensures precise joint-angle control, providing stability and robustness to uncertainties and disturbances, resulting in smoother motion and an improved user experience [87]. Data-driven models, such as deep reinforcement learning, adapt to the user’s walking style and predict gait patterns, providing personalized assistance and optimizing energy efficiency in real-time [88]. Non-model control is a superior alternative to model-based approaches as it directly manipulates control inputs based on sensor feedback [89]. Assist-as-needed control expertly adjusts the level of assistance based on the user’s needs, promoting active participation in movement tasks [6]. PID control is a proven method of controlling joint angles or forces with a perfect balance of simplicity and effectiveness [54]. Hybrid control combines multiple strategies to enhance performance, providing robustness and adaptability [90]. In the following sections, each of these approaches implemented in lower-limb exoskeletons is briefly discussed.

### 4.1. Assist-As-Needed (AAN) Control Strategies

Since exoskeletons require extensive human-computer interaction, the user’s perception needs to be greatly considered when controlling the movement of the device. Therefore, assist-as-needed (AAN) is an important control strategy that provides appropriate assistance according to the patient’s needs, leading to better results in rehabilitation and assisted walking, and is therefore receiving increasing attention from researchers of exoskeleton controllers [6,91].

The controller allows for personalized rehabilitation programs based on individual patient differences and the degree of rehabilitation progress. By dynamically adjusting the level of assistance, the patient’s rehabilitation needs can be better met, and the rehabilitation effect can be improved [92]. It is also possible to gradually reduce the level of assistance according to the patient’s rehabilitation progress, helping the patient to gradually regain the ability to walk independently. This gradual rehabilitation process can improve rehabilitation outcomes and reduce dependence on the exoskeleton [6].

By providing assistance precisely when needed, AAN approach effectively reduces the patient’s fatigue and discomfort. This helps to increase patient motivation and participation in rehabilitation and promotes rehabilitation outcomes. By improving the dynamic stability of the exoskeleton system, patients are more stable and secure during walking and movement [91]. This helps reduce the risk of accidental falls and injuries and improves rehabilitation outcomes. The AAN also monitors the patient’s movement status and rehabilitation progress in real-time and adjusts the level of assistance as needed. This real-time monitoring and adjustment ensure that the rehabilitation program is timely and effective, and improves rehabilitation outcomes [6].

### 4.2. Model-Free and Intelligent Control Strategies

Model-free control methods, such as simple Proportional-Differential (PD) or Proportional-Integral-Differential (PID) controllers, have been widely employed to ensure user stability [11,54]. However, these often fall short in addressing oscillations and underdrive issues, leading to unstable limb movements. A relatively simple adaptive PID controller can achieve a fast response through parameter adjustment and improve robustness. However, the presence of random signal noise and system disturbances, such as motor jitter, equipment weight, human jerk, sweat, and rain, can alter input and output data, adversely affecting controller performance and user safety. To address these challenges, model parameter identification methods have been proposed, utilizing statistical observation data for more accurate parameter prediction. Historically, these methods have focused on least squares, particle swarm optimization algorithms, and probability theory to compensate for the unpredictable variables affecting system performance, ensuring greater reliability and safety in controller operations [93].

The development of intelligent control algorithms, including neural networks and fuzzy logic, is deemed effective for managing system nonlinearities and disturbances. These strategies do not necessitate a mathematical model of the system but instead utilize a priori knowledge or predefined rules. For instance, neural networks incorporate preloaded system information into their architecture, whereas fuzzy control systems are governed by established rules describing system behavior. In the context of rehabilitation training, these methods are useful for estimating continuous-state parameters, such as joint angles during motion. However, continuously monitoring these parameters can be computationally intensive and pose challenges in terms of data storage. Mefoued [93] introduced a multilayer perceptron neural network for efficiently operating a lower-limb orthosis in the absence of dynamic models, addressing parameter uncertainties, as shown in Figure 6a. Peña et al. [18] utilized a multilayer perceptron neural network to process electrical muscle signals and estimate joint torque, as demonstrated in Figure 6b, showcasing potential for reducing estimation errors and improving user engagement in rehabilitation. However, the network’s inability to directly respond to muscle forces or internal parameters was highlighted. This constraint underscores the need for enhanced neural network designs or hybrid control strategies that can more effectively integrate and respond to dynamic physiological signals and internal system changes. The study suggested optimization through the torque estimated by the muscles’ electrical signals, coupled with the torque generated by the inverse dynamics tool in the OpenSim software 4.1+, to enhance user participation. However, some researchers contend that data training is computationally demanding. Hence, Pan et al. [94] proposed an innovative learning strategy with multi-loop modulation, which can analyze external disturbances while ensuring gait-learning convergence without complex computational processes. The exoskeleton’s weight may also be considered a disturbance, which can be countered by program control in addition to structural design mitigation. Yang [5] proposed an adaptive, robust control method based on servo-constraint theory and an optimized fuzzy dynamical system method. Narayan et al. [95] introduced an adaptive radial-basis-function neural network-computed torque control (ARBFNN-CTC) scheme that estimated unknown dynamics and improved gait tracking in passive-assist mode, as demonstrated through experimental trials with pediatric subjects. The ARBFNN-CTC showed significant improvements over traditional control methods and exhibited consistent performance across extended gait cycles, indicating its potential for effective long-term rehabilitation. Kenas et al. [96] developed a model-free Adaptive Nonsingular Fast Terminal Sliding Mode Controller with Super Twisting and Multi-Layer Perceptron neural network for the motion control of a 10-DOF lower limb exoskeleton in rehabilitation. The proposed controller utilized a second-order ultra-local model to simplify the exoskeleton’s complex dynamics and employed an MLP neural network to estimate the ultra-local model’s lumped disturbance. The system’s stability was analyzed using Lyapunov theory, with desired trajectories derived from surface EMG signal measurements. Co-simulation experiments conducted with MATLAB/Robotics Toolbox validated the controller’s effectiveness, showing marked enhancements in stability and precision over existing model-free controllers.

Recently, deep-learning and reinforcement-learning algorithms have also been utilized to control the movement of lower-limb exoskeletons. However, such model-free and intelligent approaches, although promising, are challenging to implement due to their time-consuming training processes and substantial hardware resource requirements. Bingjing et al. [51] suggested a reinforcement learning-based interactive control method, utilizing a double-loop nested control structure for active and passive rehabilitation. This control method combines two exoskeleton training modes, facilitating individual adaptation and active compliance in rehabilitation training [51]. Zheng et al. [88] introduced a novel strategy, SADRL, combining sensitivity amplification control (SAC) with deep reinforcement learning (DRL), to enhance lower-limb exoskeleton control. Compared to SAC alone, SADRL demonstrates superior adaptability and control effectiveness, evidenced by a significant reduction in human-exoskeleton interaction forces. However, challenges such as the need for large training datasets and the transferability of learned behaviors to real-world settings remain. Zheng et al. [89] introduced a deep reinforcement learning framework for developing a model-free walking controller for lower-limb exoskeletons, aiming to enhance human performance augmentation. The controller, based on a deep neural network, directly estimates human motion intentions without the need for a kinematic or dynamic model of the exoskeleton system. Luo et al. [43] presented a novel approach to developing a robust controller for lower-limb rehabilitation exoskeletons using deep reinforcement learning. The proposed controller is trained offline with three independent networks to handle uncertain human-exoskeleton interaction forces, enabling reliable walking assistance for users with various neuromuscular disorders. Domain randomization is employed during training to simulate patient variability, thus eliminating the need for control parameter tuning. The controller demonstrates effectiveness and robustness in virtual testing, providing assistance to users with different disabilities without specific parameter adjustments. The decoupled network structure facilitates straightforward sim-to-real transfer, showing promise for future deployment in physical systems.

### 4.3. Model-Based Control Strategies

Model-based control strategies have also been explored to address system nonlinearities. Long et al. [97] proposed a model-based control strategy, illustrated in Figure 7a, which is based on active disturbance suppression. This method extends the state observer to estimate total disturbances, followed by disturbance rejection using a predetermined method. Arnez-Paniagua et al. [26] proposed an adaptive model-based control strategy for ankle-foot orthoses that requires only a few parameters to be set, as shown in Figure 7b. These control strategies have demonstrated superior performance in trials compared to conventional PID control. Furthermore, Jeong et al. [48] developed a control method that supports the exoskeleton’s weight and maintains balance during movement. Pneumatic sensors embedded in the sole of the shoe measure the user’s foot center of pressure. Using this data, the controller for the exoskeleton was designed to provide the magnitude and direction of assistive forces through elastic actuators, effectively maintaining balance and supporting weight.

User-related disturbances are also taken into consideration. Certain rehabilitation exoskeleton devices have heavy motors and high endpoint impedance, often a result of their early manufacturing stages. Furthermore, severe spasticity in patients with neurological injuries can produce substantial response forces in the actuators, leading to unwanted device movements. To address this issue, Hussain et al. [87] proposed a model-based trajectory-tracking control strategy based on the boundary-layer-enhanced sliding-mode control law. This law facilitates guiding limb movements along physiological trajectories. The exoskeleton’s compliance in the sagittal plane was also adjusted, and the proposed control strategy was experimentally verified to effectively guide the limb through intended movements, even when structural uncertainties were present in the model [87].

### 4.4. Hybrid Control Strategies

Model-based control strategies, while precise, are often hampered by modeling inaccuracies and uncertainties, leading to potential performance degradation. Accurately modeling complex systems can be a daunting task. Conversely, non-model-based control methods rely on feedback to tailor control inputs according to the system’s actual behavior. However, these methods may fall short of achieving optimal control, particularly in reducing control effort, and the process of designing feedback control laws can be heuristic and less structured. Additionally, non-model-based control methods necessitate extensive tuning and experimentation to achieve the desired performance, particularly in complex systems. Moreover, model-free intelligent control algorithms often demand significant training or extensive datasets, which may not always be practical or achievable. Given these considerations, many studies have explored combining model-free and model-based control strategies to enhance the functionality of lower-limb exoskeletons (Figure 8).

Bao et al. [90] suggested a hybrid neuroprosthesis system for restoring lower-limb function in individuals with paraplegia, which utilized both model-based and non-model-based control methods. Specifically, a tube-based model predictive control (MPC) method was employed for model-based control to optimize coordination between functional electrical stimulation (FES) and an electric motor during knee-angle-regulation tasks. However, due to modeling uncertainties, achieving robust control performance proved challenging with this method. To address these challenges, a non-model-based control approach, incorporating nonlinear feedback control, was used to overcome model disturbances, thus enhancing the system’s robustness. Zheng et al. [89] proposed an end-to-end controller for a lower-limb exoskeleton system using deep reinforcement learning (E2EDRL). This controller consisted of high-level control for recognizing human motion intentions and low-level control for motion tracking, eliminating the reliance on complex kinematic or dynamic models. A novel multibody simulation environment and hybrid inverse-forward dynamics simulation method were introduced to facilitate safe and efficient learning of the E2EDRL strategy. The performance assessment, based on human-exoskeleton interaction (HEI) forces, demonstrated a superior control performance. Han et al. [12] suggested an intelligent PD controller based on a linear discrete-time extended state observer. This discretization effectively reduces time complexity and facilitates easier parameter tuning, achieving higher accuracy in trajectory tracking and enhanced system robustness [12].

## 5. Conclusions

This review thoroughly analyzed sensor technology for lower-limb exoskeletons, emphasizing the crucial role of sensors like EMG and EEG in monitoring physiological signals, and highlighting the importance of inertial measurement units, force sensors, and displacement sensors for orientation, interaction, and movement. The study also explored advancements in control strategies for lower-limb exoskeletons, including assist-as-needed, model-free, intelligent, model-based, and hybrid algorithms. It emphasized user safety and comfort, critically assessing the pros and cons of these approaches to offer a comprehensive perspective on the subject. Future research may likely focus on enhancing the interaction between humans and lower-limb rehabilitation exoskeletons, reducing device costs, and promoting the environmental sustainability of the materials used. It might integrate newly developed sensing technologies like vision, radar, temperature, humidity, and radiation sensors. Effective use of sensor data through control strategies could offer optimal protection for users. Anticipated advancements include integrating machine vision and human-machine interaction, necessitating more sensors and actuators and potentially leading to significant design changes. The ultimate goal is to improve the convenience and comfort of the exoskeleton for users, with simulation environments recommended for testing and validation, aiming for seamless integration into rehabilitation and better quality of life for those with lower-limb impairments.

## Figures and Tables

**Figure 1 micromachines-15-00489-f001:**
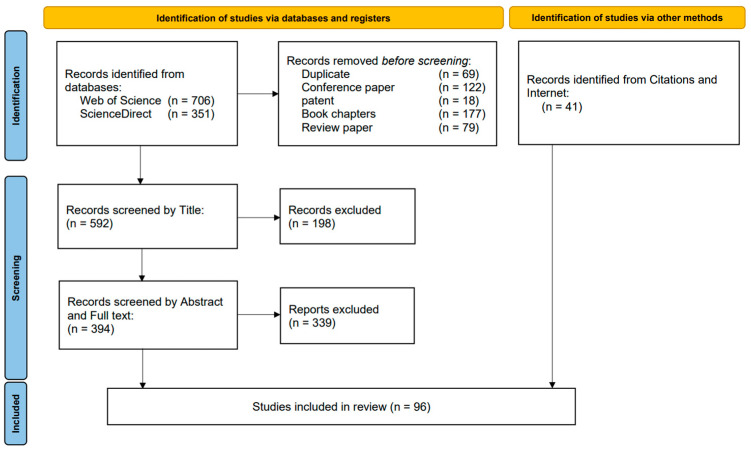
Flowchart for meta-analysis of selected articles according to preferred reporting criteria.

**Figure 2 micromachines-15-00489-f002:**
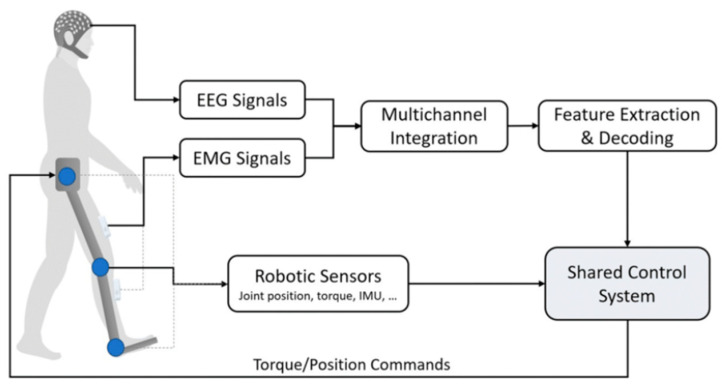
Flow architecture of signal-monitoring sensors: EMG and EEG sensors [21].

**Figure 3 micromachines-15-00489-f003:**
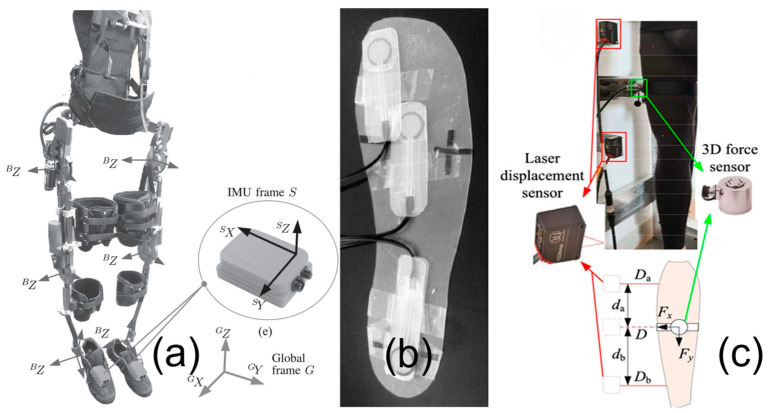
Sensors for kinematic and kinetic measurements: (**a**) IMU [25], (**b**) force sensor [26], and (**c**) displacement sensor [27].

**Figure 4 micromachines-15-00489-f004:**
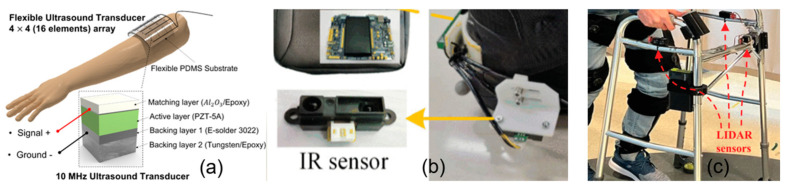
Environmental sensing used in exoskeletons: (**a**) ultrasound transducer [28], (**b**) infrared (IR) sensors [29], and (**c**) LiDAR sensors [30].

**Figure 5 micromachines-15-00489-f005:**
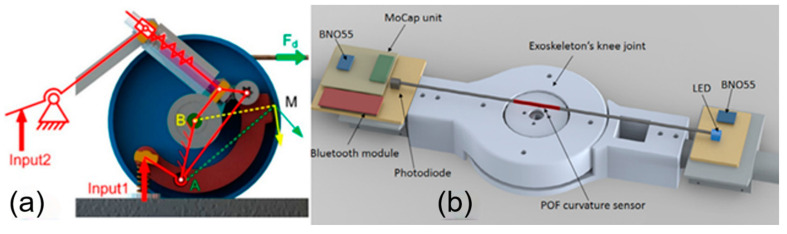
Examples of sensing systems in rehabilitation exoskeletons: (**a**) a dual-input mechanical sensing system [78], and (**b**) a knee-joint exoskeleton equipped with a Polymer Optical Fiber (POF) curvature sensor [46].

**Figure 6 micromachines-15-00489-f006:**
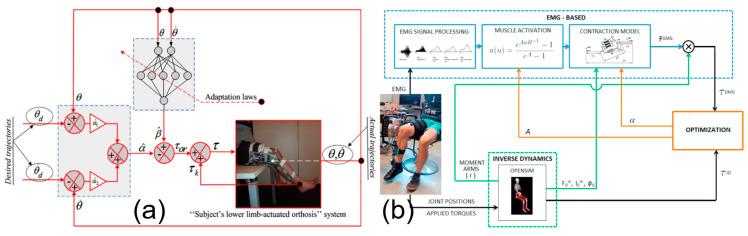
Examples of neural networks in control of rehabilitation exoskeletons: (**a**) a two-input Multilayer Perceptron Neural Network (MLPNN) [93], and (**b**) a multi-input MLPNN [18].

**Figure 7 micromachines-15-00489-f007:**
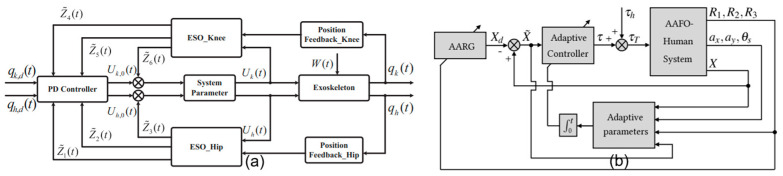
Examples of model-based control in rehabilitation exoskeletons: (**a**) a model-based control with adjustable system parameters [97], and (**b**) a model-based control with adaptive control gains [26].

**Figure 8 micromachines-15-00489-f008:**
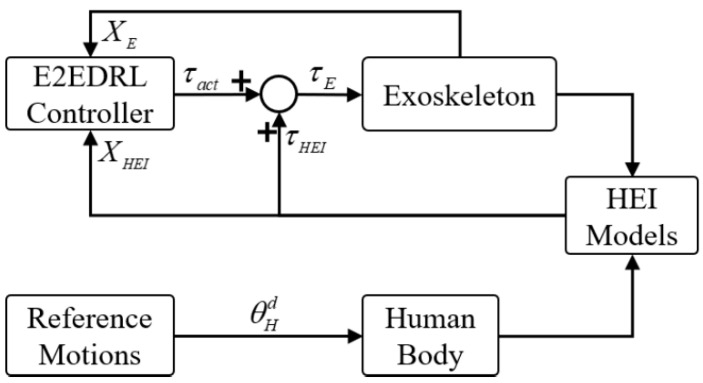
Example of a hybrid control strategy combining a kinematic/dynamic model-based controller and a non-model-based controller based on deep reinforcement learning [89].

**Table 1 micromachines-15-00489-t001:** Detailed information on the studies that use human signal monitoring sensors on lower limb rehabilitation devices.

Author, Year [x]	Sensing/Monitoring Techniques	Analyses and Control Methods	Primary Aims	Major Results and or Limitations
Kagirov et al., 2021 [3]	Position sensors and speed and acceleration sensors to control movements EMG sensors to record the work of muscles Telemetry sensors to measure temperature, rotation angles, and the current Voice-recognition technology	32-bit microcontroller STM32 MCU Intelligent remote-control interface Electric-drives control Walking-pattern control	To propose an intelligent control system with a user interface and walking mode switching for both patients and doctors.	The correctness of modeling and parameter tuning was verified by testing the exoskeleton prototype. The dual-mode control by voice and graphical interface improves the safety and convenience of use. Development of a dynamic model of the human gait process.
Chen et al., 2016 [4]	Encoders and potentiometers for receiving force feedback Rotatory potentiometers to measure the joint angle EMG sensors to estimate human intention IMU sensors to measure kinematics	Force-control strategy based on encoders and potentiometers	To design a lightweight, compact device that safely interacts with people and assists patients with gait training.	A new tandem elastic actuator, consisting of a low-stiffness spring and a high-stiffness spring, was developed and experimentally demonstrated to be effective in providing assisted force. Explore more advanced sensing and control methods and develop a new generation of prototypes with more integrated systems.
Cardona et al., 2020 [9]	EMG and IMU sensors for biomechanical model	Adaptive PD control	Quantify exercise data for personalized rehabilitation services	Movement data can be simulated by skeletal muscle models and gait improvement programs can be proposed
Li et al., 2019 [34]	EMG sensors and EEG sensors for user intention detection Inductive rotary encoders and Hall sensors to measure motor rotation angles On-board extended Kalman Filter based inertial measurement unit to measure angle information	PD controller for trajectory tracking based on hybrid EEG/EMG signals Joint feedback controller	Helping users perform daily motor tasks better.	Propose a control method based on the mixing of EEG and EMG signals from motion images and verify the effectiveness of the controller by designing a stair-climbing gait. Further improve the recognition and control algorithm.
Peña et al., 2019 [18]	EMG sensors to measure the activation of the muscles EMG-driven torque-estimation method	Multilayer perceptron neural network Online adaptive impedance controller	To estimate the appropriate torque and optimal impedance control for the user during the wearing of the exoskeleton.	Torque is estimated and optimized using a simplified model with a specific EMG signal, and the experiments show that the use of the EMG signal is very effective for adaptive control strategies. To try out the exoskeleton on patients with limb injuries and to verify its effectiveness.
Tu et al., 2022 [35]	IMU sensors to measure the angles Insole-type plantar pressure diaphragm sensors to measure ground reaction force EMG sensors	User intention detection Active height-adjustment control	To protect the user’s body during squatting activities.	The new E-LEG system reduces the user’s muscle activity during squatting and effectively relieves muscle strain.
Huang et al., 2019 [36]	EMG sensors placed on the legs and hips Load-cell force sensors for precise data Pedal force sensors to measure the human-pedal interaction forces	Sensor-fusion method PD feedback controller	Helping patients with lower limb movement disorders to perform safe rehabilitation exercises.	Prototype demonstrates the effectiveness of muscle exercise through EMG signals in an experiment. Further confirm the feasibility of the design and procedure, and involve people with different levels of health in the experiment.
Choi et al., 2020 [37]	Pressure sensors and resistive force sensors to measure the gait cycle Tensile load cell sensor to measure the output of the pneumatic artificial muscles Surface EMG signals in the experiment process	Pulse-width modulation (PWM) signals ADC function of the microcontroller named STM32F407VG Sliding mode control	To design an ankle-foot orthosis that generates enough assisted force to aid the user’s ankle motion	Using tandem elastic actuators and polylactic acid material to enhance human ankle-joint strength, device performance was tracked and tested Control the balance of the device in any state by controlling the torque, etc.
He et al., 2019 [38]	Flexible piezoresistive force sensors to measure one-dimensional forces Six-axis IMU sensors to detect the movement Encoder and torque sensors to drive the actuators EEG, EMG sensors for human-robot interaction signals CAN bus EtherCAT system	STM32F103 microprocessor Bipedal walking algorithms Self-balanced walking algorithms on flat terrain	To propose an anthropomorphic exoskeleton design and control.	A supervised algorithm was developed to detect synchronized movements of the user and the exoskeleton, and a significant reduction in muscle activity was measured. Improve the detection accuracy of the filter and try other methods to provide assisted force to the user.
Hassani et al., 2014 [39]	EMG sensors to measure the muscular activities incremental encoders to measure the joint angles	The Dspace DS1103 controller board and the Maxon motion controller EPOS 70/10 Phase-adaptive control framework	To propose an interactive method of active and passive limb movements that can achieve repetitive limb movements and intention-driven limb movements, respectively.	Satisfactory results were obtained in many aspects through experiments with healthy people. Propose a control framework that can help users to perform different tasks such as combat and stair climbing.
Yi et al., 2022 [40]	Nine-axis IMUs embedded in surface electrodes (Delsys Trigno) to measure the joint angles 8-camera video system to record the 3-D locations EMG sensors to measure the muscle activities	EMG-IMU sensor fusion Ahead-of-time prediction algorithm	To use the latest technology to detect the user’s intention to move and to avoid delays caused by transmission.	The proposed over-the-top continuous prediction method is tested on the knee joint and outperforms the traditional method. The actual usage environment is very diverse and there is a lack of algorithms for different movement patterns.
Bayon et al., 2022 [41]	Resistive force sensors insole to measure the ground reaction force EMG sensors to measure muscle surface activities. IMU sensors to estimate body center-of-mass	Balance assistive controller Perturbation detection algorithm	To help users maintain balance when standing or moving.	A collaborative ankle-ankle exoskeleton control strategy is proposed to effectively reduce muscle activity and maintain balance control in the experiment. Extension of the control strategy to multi-joint exoskeleton systems.
Jradi et al., 2024 [42]	Three force-sensitive resistor was used to measure the ground reaction force Two EMG sensors and two IMU sensors	An adaptive active disturbance rejection controller	Ankle-foot orthoses are utilized to provide continuous assistance to patients with foot drop while walking and to supplement the user’s muscle strength.	The proposed adaptive active-interference-suppression controller is adaptive with enhanced synchronized motion of the ankle joint. The joint motion may affect the neighboring joint motion, and the effect of the controller will be verified on more clinical patients in the future.
Luo et al., 2023 [43]	Load-cell force sensors EMG sensors	DAQ card linked to mini-PC Microprocessor ARM CORTEX-M4	Human-computer systems trained by simulation to predict user HCI forces.	Proposes decoupled RL-based control, which can be trained in arbitrary situations to avoid discomfort during human-computer interaction.

**Table 2 micromachines-15-00489-t002:** Detailed information of studies that use IMUs, force, and displacement sensors to measure kinematic and kinetic of human-lower limb rehabilitation devices.

Author, Year [x]	Sensing/Monitoring Techniques	Analyses and Control Methods	Primary Aims	Major Results and or Limitations
Zhou et al., 2020 [1]	Ultra-small rotary magnetic encoder sensors Cable-displacement force/torque sensors (MINI 45, ATI)	Assess the gravity compensation by perceived assistance	A passive lower-limb exoskeleton using a spring structure is proposed for assisted walking.	A prototype of the exoskeleton has been built and the results of tests on healthy people have verified the usability of the exoskeleton. Verification of the user’s lumbar force and quantitative evaluation of the EMG signal acquisition.
Chen et al., 2016 [4]	Encoders and potentiometers for receiving force feedback Rotatory potentiometers to measure the joint angle EMG sensors to estimate human intention IMU sensors to measure kinematics	Force-control strategy based on encoders and potentiometers	To design a lightweight, compact device that safely interacts with people and assists patients with gait training.	A new tandem elastic actuator, consisting of a low-stiffness spring and a high-stiffness spring, was developed and experimentally demonstrated to be effective in providing assisted force. Explore more advanced sensing and control methods and develop a new generation of prototypes with more integrated systems.
Sado et al., 2019 [7]	Position sensors to measure the joint angle Resistive force sensors to measure the ground reaction force	Dual extended Kalman filter sensor-less joint torque estimation Linear quadratic Gaussian torque amplification controller Supervisory controller	To propose an anthropomorphic exoskeleton design and control.	A supervised algorithm was developed to detect synchronized movements of the user and the exoskeleton, and a significant reduction in muscle activity was measured. Improve the detection accuracy of the filter and try other methods to provide assisted force to the user.
Cardona et al., 2020 [9]	EMG sensors and IMU sensors for biomechanical model	Adaptive PD-control strategy EPOS2 70/10 digital position controller	To quantify patients’ exercise data and provide personalized rehabilitation services.	Patients’ movement data can be simulated by human skeletal muscle models and gait improvement programs can be proposed. Assemble and fabricate the exoskeleton prototype, and verify the stability and feasibility of the prototype with engineering knowledge.
Han et al., 2020 [12]	Force sensors to measure interaction force and foot pressure Maxon Encoder (Mile 1024 CPT) to measure the angle position and velocity	A linear discrete-time extended state observer based intelligent-PD controller	A novel tracking differential controller is designed and a hyperlocal model is used to obtain real-time velocity and acceleration.	Experimental and simulation validation methods are used to verify the higher performance of the controller, motor, and other hardware devices.
Huo et al., 2018 [25]	Resistive force sensors to measure the ground reaction force IMU sensors to obtain the orientation rotation matrix	Sensor fusion with Kalman Filter Gait-mode-detection-based torque assistive controller	To propose a sensor-based approach for fast gait detection.	Real-time gait estimation using fuzzy logic algorithms allows the selection of the appropriate kinetic and kinematic model for each gait, but currently it is still not applicable to walking at too low a speed. Personalized and customized assist algorithm, trying to increase the output power to achieve higher assist ratio.
Arnez-Paniagua et al., 2019 [26]	IMU sensors to measure the acceleration Incremental encoders to measure the joint angle Resistive force sensors to measure the ground reaction force	Gait-cycle detection Gait-sub-phases detection by a Mamdani fuzzy inference system Adaptive ankle reference generator algorithm	To control orthotics to help patients with gait impairment to walk normally.	A model-based control method is proposed for the control of dorsiflexion and supination movements of the ankle joint.
Li et al., 2017 [27]	Force sensors (ATI) to measure the ground reaction force Rotary encoders to measure the spring displacement Rotary potentiometers to measure the joint angle	The Dspace control system Multi-modal control scheme (consists of three control modes: robot-assisted, robot-dominant, and safety-stop mode)	Addressing instability of use due to nonlinear factors.	A control scheme combining three modes is proposed to achieve on-demand assistance, corrective motion, and stop motion in different ranges, respectively. Do more experiments in clinical and rehabilitation areas.
Leal-Junior et al., 2018 [46]	IMU sensors Polymer optical fiber (POF) curvature sensors (a kind of displacement sensor) Hysteresis compensation technique	Tree analysis of current problems Sensor-fusion algorithms Verification of algorithm validity using root-mean-square error	Breaking through the strain limit and fracture toughness of ordinary curvature sensors, the softness of fibers is similar in order of magnitude to that of the human body.	Fusing two sensors to complement each other’s defects and improve the measurement accuracy of the sensors compared to not using an exoskeleton. To make a basis for future research on sensor-fusion methods and to replace other traditional sensors with lower costs.
Jeong et al., 2020 [48]	Air pressure sensors embedded in the shoes EMG sensors for muscle surface activities IMU sensors for inclination angles	Feedback controller Feedforward controller Weight support and balance control method	Maintains better stability during user movement and provides partial weight support.	A new control method using air-pressure sensors to detect the user’s center of gravity in real time is proposed, assorting a series of elastic actuators to generate corresponding auxiliary torque. Weakened signals measured by EMG sensors verify the effectiveness of the device on assist the human body. The experiment should be extended to more people and observe the changes in their other physical indicators.
Zhang et al., 2018 [49]	Absolute position magnetic encoders (MBA8) to measure joint angles Torque sensors to measure the interaction torque IMU sensors (VN-100S) to estimate the extrapolated center of mass Resistive pressure sensors (FSR 402) to detect the ground contact	Admittance-based controller Balance controller	To develop an active compliant exoskeleton for the hip joint that can assist users in multiple planes of motion.	The proposed balance control strategy generates enough guiding forces. Preliminary tests on healthy subjects demonstrated the effectiveness. Expand the subject population to make it effective in clinical and geriatric populations, as well as consider balance problems under other perturbations.
Taherifar et al., 2018 [50]	Single-axis tension-compression force sensors to measure the interaction forces Angular position sensors to measure the position errors	Assist-as-needed controller based on the impedance control Human swing leg control	To design an exoskeletal rehabilitation system that provides assistance as needed. Based on the target impedance gain and feedforward force, a parameter was set to assist in determining whether assistance is needed.	The system was able to steadily reduce the force during human-machine interaction, and the use of a tandem elastic actuator was shown to contribute to impedance control.
Bingjing et al., 2019 [51]	Linear potentiometer sensors to measure the joint angles Resistive force sensors to measure the human-robot interaction force	Position controller Reinforcement-learning interactive controller Velocity feedforward algorithm Gravity compensator	To propose a new human-machine interaction control strategy to ensure gait tracking accuracy.	An adaptive control strategy based on sigmoid functions and reinforcement learning algorithms, combined with flexible pneumatic actuators, is used to finally verify the effectiveness of the strategy. Implementing active resistance rehabilitation training.
Aguirre-Ollinger and Yu, 2020 [52]	Actuator encoders, linear potentiometers, knee joint encoders, and IMUs to measure the kinematic data Rotary encoders to measure the angular positions	Force tracking feedback controller based on the forward-propagating Riccati equation	Increasing user engagement with traditional tandem elastic actuators.	Propose variable structure tandem elastic actuator with adjustable stiffness according to different commands. Interference suppression components are used to increase transparency in the zero-force case, and force control is used to correct partially asymmetric gaits.
Chen et al., 2015 [53]	IMU sensors and pressure sensors in both shoes for locomotion mode recognition	Gait-event detection Parameter optimization-based neural-machine-interface control strategy	To propose a better strategy to classify the motion characteristics in different phases and improve the parameters such as feature set and window size.	The inertial and pressure feature sets can be measured through different motion tasks of the subjects, thus providing better recognition performance. To make the system more integrated and to further improve the recognition performance.
Gasparri et al., 2019 [54]	High-range force sensors (FlexiForce A201, Tekscan) to measure the ground reaction force Torque sensors to track the instantaneous torque profile	Proportional joint-moment controller 32-bit ARM microprocessor	Assist users in maintaining stability in different sports and improve clinical usability.	The ankle exoskeleton using joint-torque-control strategy is developed, and the control algorithm is also designed to adapt to different motion decisions of users. Evaluate whether the metabolic cost of users is reduced and test it on a larger scale.
Peña et al., 2019 [18]	EMG sensors to measure the activation of the muscles EMG-driven torque-estimation method	Multilayer perceptron neural network Online adaptive impedance controller	To estimate the appropriate torque and optimal impedance control for the user during the wearing of the exoskeleton.	Torque is estimated and optimized using a simplified model with a specific EMG signal, and the experiments show that the use of the EMG signal is very effective for adaptive control strategies. To try out the exoskeleton on patients with limb injuries and to verify its effectiveness.
Tu et al., 2022 [35]	IMU sensors to measure the angles Insole-type plantar pressure diaphragm sensors to measure ground reaction force EMG sensors	User-intention detection Active height-adjustment control	To protect the user’s body during squatting activities.	The new E-LEG system reduces the user’s muscle activity during squatting and effectively relieves muscle strain.
Kim and Cho., 2019 [55]	Six-axis force/torque (FT) sensors to detect user intention CAN bus	Balancing controller z-directional admittance controller vibration reduction algorithm and soft-landing algorithm for swing leg compliant control algorithm for upper body moving stop algorithms for swing leg and upper body	Assist users to complete the sit-to-stand transition.	A new frame is proposed: KUEX, worn on the anterior side of the user, which can effectively reduce the wearing time. There are difficulties in turning movements.
Huang et al., 2019 [36]	EMG sensors placed on the legs and hips Load-cell force sensors for precise data Pedal force sensors to measure the human-pedal interaction forces	Sensor-fusion method PD feedback controller	Helping patients with lower-limb movement disorders to perform safe rehabilitation exercises.	Prototype demonstrates the effectiveness of muscle exercise through EMG signals in an experiment. Further confirm the feasibility of the design and procedure, and involve people with different levels of health in the experiment.
Li and Hashimoto, 2016 [37]	Resistive strain-gauge force sensors under the feet to detect gait cycles PVC gel soft actuators include a laser displacement sensor (IL-065, Keyence) myRIO (LabVIEW) computer for real-time data acquisition and conversion	Tree analysis of the soft actuator Gait-cycle detection Operation decision control DC field control	To design an advanced, easy-to-wear soft plastic gel actuator to assist users in walking.	The actuator was shown to reduce muscle load and make walking more efficient in life, and to prevent actuator breakage. Research on building an effective control algorithm for this prototype exoskeleton, designing an intelligent battery, and reducing the operating voltage.
He et al., 2019 [38]	Flexible piezoresistive force sensors to measure one-dimensional forces Six-axis IMU sensors to detect the movement Encoder and torque sensors to drive the actuators EEG, EMG sensors for human-robot interaction signals CAN bus EtherCAT system	STM32F103 microprocessor Bipedal walking algorithms Self-balanced walking algorithms on flat terrain	To propose an anthropomorphic exoskeleton design and control.	A supervised algorithm was developed to detect synchronized movements of the user and the exoskeleton, and a significant reduction in muscle activity was measured. Improve the detection accuracy of the filter and try other methods to provide assisted force to the user.
Bayon et al., 2022 [41]	Resistive force sensors insole to measure the ground reaction force EMG sensors to measure the muscle surface activities IMU sensors to estimate body center-of-mass	Balance assistive controller Perturbation detection algorithm	To help users maintain balance when standing or moving.	A collaborative ankle-ankle exoskeleton control strategy is proposed to effectively reduce muscle activity and maintain balance control in the experiment. Extension of the control strategy to multi-joint exoskeleton systems.
Park et al.,2021 [56]	IMU sensors (3DM-GX4-25) to measure the torso orientations Compact torque Sensors (TQ-CSKG02-NM150) to measure joint torques Absolute encoders (RMB28SC) to measure joint angles; multi-turn absolute encoders (EBI1135) to measure the actuator positions. Resistive force sensors to measure the ground reaction forces	Pulley controller using the system dynamics	To reduce the mass and moment of inertia of the exoskeleton to extend the service time.	Propose a cable differential mechanism to provide sufficient torque and speed. To validate the effectiveness of the prototype on patients’ lower limb motor rehabilitation ability and metabolic exertion.
Huang et al., 2018 [57]	Encoders to measure the current state of HUALEX system IMU sensors to measure the walking velocity Plantar sensors to judge the current phase	Hierarchical interactive learning controller (motion learning and model-based controller learning) Node controller set nearby each active joint	To propose a control strategy with a hierarchical interaction learning framework that can handle different human-computer interaction movements.	Experimental results show that the control algorithm has more processing power and better performance. An attempt is made to use a soft modelling approach to guide the motion of the exoskeleton with sensitivity coefficients.
Shi et al., 2021 [58]	EtherCAT system (Angle sensors to measure the terminal posture, encoder of the motor to measure the speed, IMU sensors to measure the actual hip angle)	Model-based human-centered adaptive controller	To propose a human-centered interaction control method to mitigate the errors caused by band connections.	The adaptive controller is designed based on the dynamic model of human-computer interaction and its effectiveness is verified by simulation.
Hwang et al., 2019 [59]	IMU sensors to measure the movement intention Kinect sensors to obtain the joint position Encoder sensors to measure the joint angle	Trajectory tracking control method	Development of exoskeleton devices based on the actual needs of the user such as gait cycle.	The average value of gait data of ordinary people and the data obtained from Kinect sensor predefined the exoskeleton gait and verified the effectiveness of the learning algorithm.
Foroutannia et al., 2022 [23]	Resistive force sensors to measure the ground reaction force IMU sensors to measure the speed and body position CAN bus Wet Ag/AgCl electrodes-based EMG sensors to measure the muscle activities Small beam-type load cells to measure the interaction force between the human-robot system	BECKHOFF programmable logic controller	To predict joint positions more accurately.	The effectiveness and stability of the algorithm was experimentally verified using an EMG-based deep learning neural network placed in an impedance control loop. Algorithm fusion strategies should be used to develop more disease-specific controllers.
Matinez-Hernandez et al., 2022 [60]	Nine-axis IMU sensors to measure the angular velocity, accelerometer, and magnetometer signals Insole resistive force sensors to detect the gait cycle	Sensor fusion Convolutional Neural Network (CNN) Gait cycle tracking Predicted Information Gain method	To predict gait cycles more accurately and to identify walking activity.	An adaptive strategy combining convolutional neural networks and predictive information gain is proposed and its accuracy is experimentally verified.
Wang et al., 2021 [61]	Encoders to measure the piston position in the valve Force sensors to measure the pressure at the outlets of the valve Displacement sensors to measure the real joint angle	Piecewise PID controller	The motion and drive of the exoskeleton are optimized to improve the tracking accuracy and assist performance.	A control method with error estimation and compensation is proposed, and the feasibility is verified by simulation.
Long et al., 2017 [62]	Digital pressure sensors to measure the interaction force between ground and exoskeleton Optical encoders to measure the angular position Gasbag-based force sensors to measure the interaction force between human limbs and the exoskeleton limbs CAN bus	Sensor fusion online Walking phase identification Adaptive minimizing pHRI control strategy Static balance control in the stance phase	Improving comfort and reducing energy consumption when wearing an exoskeleton	A sensor-based adaptive control strategy is proposed to continuously keep the human-machine interaction force at a minimum. This strategy can effectively assist walking by verifying the mean-square value of human-machine interaction force and torque force. To Improve applicability for different users.
Urendes et al., 2019 [63]	Potentiometer sensors to measure the joint angle Resistive force sensors in each shoe Ultrasound sensors measure the distance for synchro the whole system CAN bus	PWM controller Microcontroller board dsPIC30F4013 Weight support control	Design of a system called HYBRID for monitoring and analyzing user limb movement data.	Incorporating features such as gait guidance and weight support, it uses a newer human-computer interaction solution that eliminates the need for cables and allows for appropriate patient proprioceptive stimulation during movement.
Cestari et al., 2015 [64]	Embedded force sensors to detect small torques Industrial CAN-OPEN bus	Maxon EPOS controllers and NI CompacRIO controllers	Reducing the energy requirements of compliant actuators and enhancing adaptability to external disturbances	An adjustable-rigidity and embedded sensor joint prototype for children was designed, and the proposed rigidity-adjustable design combined with different force control strategies can effectively achieve energy savings in the device.
Li et al., 2022 [65]	Load cells (LSB201, FUTEK) to measure the tension force Insole resistive force sensors for switching the gait	Adaptive position tracking controller Impedance learning and hierarchical controller	To consider impedance matching and environmental factors while meeting flexibility requirements.	To propose an adaptive control strategy based on impedance learning and considering users. Limitations: Only for a single joint.
Ashmi and Akhil, 2023 [66]	Pressure sensors Accelerometer ADXL335 Optical sensor	ATmega328 microcontroller PID controller using particle swarm optimization	This paper notes the importance of controllers and designs suitable controllers for the knee and hip joints of the exoskeleton.	The PID control in this paper exhibits less oscillation and better steady state error compared to other controllers. The global optimum position can be obtained by finding the optimum controller gain constant using particle swarm optimization.
Zhong et al., 2023 [67]	Instep-mounted IMUs and encoders EMG sensors	Microcontroller (STM32F407) Upper controller (Raspberry CM4)	Improving personalized gait performance and gait symmetry for stroke patients.	A hybrid cable-actuated exoskeleton was developed and was positively received by patients who used it. The extent of the efficacy of the rehabilitation training remains to be determined, and a larger sample and time will be used in the future.

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
