# Peer review of "Advancements in Sensor Technologies and Control Strategies for Lower-Limb Rehabilitation Exoskeletons: A Comprehensive Review"

_micromachines, 2024, doi:10.3390/mi15040489_

Round 1

Reviewer 1 Report

Comments and Suggestions for Authors

The authors systematically review lower limb rehabilitation exoskeletons, evaluating their sensor technologies, control strategies, and structural designs. They assess various sensing modalities and control algorithms to improve exoskeleton performance and enhance the quality of life for patients with mobility impairments. However, the manuscript could benefit from several revisions to enhance its clarity and depth.

1. The introduction section lacks clarity regarding how this review differs from previous studies. The authors are encouraged to dedicate a separate paragraph to delineate the distinctions between their work and existing literature, such as the  [10.1016/j.arcontrol.2022.04.003], [10.3390/machines11070764] and so on.

2. In Methodology, the authors should employ a proper PRISMA flowchart to illustrate the detailed screening of relevant articles. Additionally, they should expand the keyword selection to include terms like "lower-extremity exoskeleton" and "gait rehabilitation." Grouping keywords with appropriate conjunctions like "or" and "and" can help avoid redundancy.

3. Figures and tables need revision. Figure 2 should depict the flow architecture of multimodal sensors used in exoskeleton systems. Tables 1 and 2 require updates to include recent references on exoskeleton control, particularly from the last 3-4 years. Examples of relevant studies are provided, such as [10.1186/s12984-023-01147-2], [10.1016/j.medengphy.2023.104080], and [10.3390/bioengineering11020188].

4. The control section of the paper needs significant improvement. It is suggested to restructure it with subsections focusing on different control modalities, (such as model-based, model-free, hybrid in position control, impedance/admittance control in human-in-the-loop control, and supervisory control such as EEG, EMG, and so on). Recent works on intelligent control schemes, like [10.1007/s11370-023-00477-3] and [10.1002/acs.3723], should be cited. Each subsection should discuss the advantages and limitations of existing control methods for lower limb exoskeletons, supplemented with schematic diagrams of control architectures.

5. A separate section on clinical benefits and progress post-sensor, actuator, and control scheme implementation is needed. It should elucidate how various sensing modalities, such as electromyography (EMG), contribute to motion control accuracy and responsiveness in exoskeletons. Furthermore, the distribution of sensors within exoskeletons and their impact on structural design and effectiveness should be discussed.

6. What are some of the latest control algorithms and analysis methods to optimize exoskeleton performance and ensure safe user interactions? Surprisingly the authors have hardly cited any paper on 'assist-as-need' control which is quite important for safe and robust human-exoskeleton interactions.

7. The interplay between control algorithms and sensor systems, as well as measures ensuring user safety and effectiveness, needs further exploration. Specific attention should be paid to how these elements enhance exoskeleton functionality and user experience.

Reviewer 2 Report

Comments and Suggestions for Authors

The scientific relevance of the article would be enhanced by extending in the Discussion a summary of the limitations of the research/publications to date and highlighting the main directions for further research.

Reviewer 3 Report

Comments and Suggestions for Authors

The study of Yao and Colleagues aim to reviews the state of the art in lower limb exoskeletons designed to assist people with movement disorders, mainly analyzing papers up to 2021, even if Authros also discussed papers up to 2023. The paper is well written and the organization of the work is clear with an excellent view on the state of the art. I have only few main points that I listed below.

Main Points

1)      Why do the Authors limited the paper in Table up to 2022?  Since Authors have done an excellent work I would suggest to go further and frame up to the latest issues of 2024. In any case this is demanded to the Authors.

2)      Regarding the review of the control approaches, I appreciated the subdivision in gray, black and white box. However since gray and  black requires actually system identification procedures or the development of data-driven models based on machine or deep learning I suggest to point out which part of the control scheme is modeled through this approach. Indeed, such data driven models can be inserted in more large control scheme for modelling the human-machine interaction or for predicting the gait phases or the joint angles. This is important since the final exo can have lower and higher level control schemes orchestrated in a hierarchical way. Authors can help the readers also by introducing a graph or graphs, as done for figure 6 and 7 but with a generalizing purpose.

3)      When Authors discuss about EMG driven models and thus exo, it is important to point out the pros and cons of using EMG. Indeed the EMG signal anticipates the movement and thus carries the information of human movement volition and intention, this can be use as trigger but also for developing more advanced models. I suggest the Authors to report and discuss this point based also on the following paper:

-"A Minimal and Multi-Source Recording Setup for Ankle Joint Kinematics Estimation During Walking using only Proximal Information from Lower Limb." IEEE Transactions on Neural Systems and Rehabilitation Engineering (2024).

I suggest the Authors to stress  the importance of combining kinetic data with physiological data as EMG and EEG, also discussing in a proper section the papers that combined the information and the aims for which these information are combined. Although this aspect seems secondary with respect to the narrative of the sensors employed, I believe that smart sensors for exo’s unavoidably have to be designed considering the perspective of dealing with a very complex phenomenon i. e. the interaction of  a human and a machine for mitigating impairments. 

Round 2

Reviewer 1 Report

Comments and Suggestions for Authors

The authors have addressed most of the concerns raised by the reviewer. However, still there is a lack of comparison between current review work and similar studies (authors should compare with the articles given in first comment in first revision)- there are many similar works in the literature. 

Author Response

Please find the revisions in red color at the near end of the introduction. We addressed the previous similar reviews and emphasized the merits of our research.

Reviewer 3 Report

Comments and Suggestions for Authors

Authors responded to all my concerns. This review can be beneficial for the community and deserves to be published.

Author Response

Thank you so much for your review